# Dose-dependent phosphorylation and activation of Hh pathway transcription factors

Mengmeng Zhou[1],*, Yuhong Han[1],*, Bing Wang[1], Yong Suk Cho[1], Jin Jiang[1,2]

**Graded Hedgehog (Hh) signaling is mediated by graded Cubitus interruptus (Ci)/Gli transcriptional activity, but how the Hh gradient is converted into the Ci/Gli activity gradient remains poorly understood. Here, we show that graded Hh induces a progressive increase in Ci phosphorylation at multiple Fused (Fu)/CK1 sites including a cluster located in the C-terminal Sufu-binding domain. We demonstrated that Fu directly phosphorylated Ci on S1382, priming CK1 phosphorylation on adjacent sites, and that Fu/CK1-mediated phosphorylation of the C-terminal sites interfered with Sufu binding and facilitated Ci activation. Phosphorylation at the N-terminal, middle, and C-terminal Fu/CK1 sites occurred independently of one another and each increased progressively in response to increasing levels of Hh or increasing amounts of Hh exposure time. Increasing the number of phospho-mimetic mutations of Fu/CK1 sites resulted in progressively increased Ci activation by alleviating Sufu-mediated inhibition. We found that the C-terminal Fu/CK1 phosphorylation cluster is conserved in Gli2 and contributes to its dose-dependent activation. Our study suggests that the Hh signaling gradient is translated into a Ci/Gli phosphorylation gradient that activates Ci/Gli by gradually releasing Sufu-mediated inhibition.**

## Introduction

The Hedgehog (Hh) family of secreted proteins plays critical roles in embryonic development, adult tissue homeostasis and regeneration, and cancer progression (Taipale & Beachy, 2001; Jiang & Hui, 2008; Briscoe & Therond, 2013; Jiang, 2021). Initially identified as an oncogene amplified in glioma, the Gli family of transcription factors transduce the Hh signal to the nucleus to regulate Hh pathway target genes involved in the regulation of cell growth and patterning in species ranging from *Drosophila* to human (Hui & Angers, 2011). Gli and its *Drosophila* homolog Cubitus interruptus (Ci) exist in two distinct forms. In the absence of Hh, full-length Gli/Ci (Gli$^F$/Ci$^F$) is proteolytically processed into a truncated repressor form

(Gli$^R$/Ci$^R$), which lacks the C-terminal coactivator–binding domain and actively represses the expression of a subset of Hh target genes (Aza-Blanc et al, 1997; Wang et al, 2000a). In the presence of Hh, Gli/Ci processing is blocked, leading to de-repression of Hh target genes. In addition, Hh stimulates the activity of full-length Gli/Ci and converts it into a transcriptionally activator form (Gli$^A$/Ci$^A$) that enters the nucleus to activate Hh target genes (Methot & Basler, 1999; Aza-Blanc et al, 2000).

A salient feature of Hh signaling is that different levels of Hh ligand trigger different developmental outcomes. It is thought that different levels of Hh signaling activity is mediated by two reversed gradients of Gli$^R$/Ci$^R$ and Gli$^A$/Ci$^A$ with increasing levels of Hh progressively decrease the levels of Gli$^R$/Ci$^R$ activity but increase the levels of Gli$^A$/Ci$^A$ activity. Much is known about how Gli$^R$/Ci$^R$ is regulated. The absence of Hh allows Gli$^F$/Ci$^F$ to be phosphorylated by multiple kinases including PKA, glycogen synthase kinase 3 (GSK3), and casein kinase 1 (CK1) (Jiang & Struhl, 1995; Wang et al, 1999; Wang et al, 2000a; Jia et al, 2002; Price & Kalderon, 2002), which targets it for ubiquitination by an E3 ubiquitin ligase SCF$^{Slimb/\beta\text{-TRCP}}$, followed by proteasome-mediated partial degradation to generate Gli$^R$/Ci$^R$ (Jiang & Struhl, 1998; Jia et al, 2005; Jiang, 2006; Tempe et al, 2006; Smelkinson et al, 2007). In *Drosophila*, Ci phosphorylation is regulated by a protein complex containing a kinesin-like protein Costal2 (Cos2) and a Ser/Thr kinase Fused (Fu), which forms a complex with Ci and its kinases including PKA, GSK3, and CK1 to facilitate Ci phosphorylation and processing (Zhang et al, 2005). Hh signaling releases Ci and its kinases from the Cos2/Fu complex, thereby inhibiting Ci phosphorylation and processing (Zhang et al, 2005; Li et al, 2014; Ranieri et al, 2014). In mammalian cells, Hh signaling is thought to regulate the local activity of PKA in the primary cilium to inhibit Gli processing (Mukhopadhyay et al, 2013; Arveseth et al, 2021; Truong et al, 2021).

Compared with the regulation of Gli$^R$/Ci$^R$, how Hh signaling converts Gli$^F$/Ci$^F$ into Gli$^A$/Ci$^A$ is less understood (Zhou & Jiang, 2022). In *Drosophila*, Fu is required for converting Ci$^F$ into a labile Ci$^A$ by alleviating the inhibition by Sufu, a conserved pathway regulator that binds and inhibits Ci/Gli (Ohlmeyer & Kalderon, 1998; Methot & Basler, 2000; Wang et al, 2000b; Humke et al, 2010; Tukachinsky et al,

[1]Department of Molecular Biology, University of Texas Southwestern Medical Center, Dallas, TX, USA   [2]Department of Pharmacology, University of Texas Southwestern Medical Center, Dallas, TX, USA

Correspondence: jin.jiang@utsouthwestern.edu
*Mengmeng Zhou and Yuhong Han contributed equally to this work.

2010; Zhang et al, 2013; Shi et al, 2014a; Han et al, 2015). Although Hh induced Sufu phosphorylation in a manner depending on Fu (Lum et al, 2003), preventing Fu-mediated phosphorylation of Sufu by mutating the relevant sites did not affect Fu-mediated activation of Ci either in vitro or in vivo (Zhou & Kalderon, 2011; Han et al, 2019). Our recent study demonstrated that Ci is a direct and physiologically relevant substrate of Fu (Han et al, 2019). We found that Fu directly phosphorylated Ci on Ser218 and Ser1230, which primed its further phosphorylation by CK1 on adjacent sties and that these phosphorylation events promoted Ci activation by attenuating Sufu/Ci interaction. Furthermore, we found that Shh stimulated phosphorylation on a conserved cluster of Ser residues in the N-terminal region of Gli proteins through the Fu-related kinases Ulk3 and Stk36 (Han et al, 2019).

In the present study, we identified a new Fu phosphorylation site located in the C-terminal Sufu-binding domain of Ci. Using in vitro kinase assay and phospho-specific antibodies, we demonstrated that Fu phosphorylated Ci on S1382 in response to Hh, which primed its further phosphorylation by CK1 on adjacent sites. We found that Fu/CK1–mediated phosphorylation at this phosphorylation cluster interfered with Sufu binding and modulated Ci activity in conjunction with other phosphorylation clusters. We showed that phosphorylation at individual Fu/CK1 sites occurs independently of one another, but each increased progressively in response to increasing levels of Hh ligand or increasing amounts of Hh exposure time. We provide evidence that increasing levels of Ci phosphorylation at Fu/CK1 sites resulted in a progressive increase in $Ci^A$ activity. We also found that a similar phosphorylation cluster in the C-terminal Sufu-binding domain of Gli2 contributed to its dose-dependent activation. Taken together, our results suggest that the Hh activity gradient is translated into a Gli/Ci phosphorylation gradient that fine tunes $Ci^A$/$Gli^A$ activity.

# Results

### Fu phosphorylates the C-terminal Sufu-binding domain of Ci

Our previous study uncovered a Sufu-binding site in the C-terminal region (SIC) of Ci from aa1371 to aa1397, which is conserved in both Gli2 and Gli3 (Han et al, 2015). Inspection of this region revealed that Ser 1382 (S1382) falls into Fu phosphorylation consensus site: $S/TX_5E/D$ (Fig 1A) (Han et al, 2019). To determine whether Fu could directly phosphorylate Ci on S1382, we carried out an in vitro kinase assay using GST fusion proteins containing the Ci C-terminal fragment (GST-CiC; Fig 1B) as substrates and Flag-tagged constitutively-active (CC-Fu$^{EE}$) or kinase-dead (CC-Fu$^{KR}$) Fu purified from insect cells as the kinase source (Shi et al, 2011; Han et al, 2019). Phosphorylation of GST fusion proteins was detected by the non-radioactive method (pIMAGO) (Han & Jiang, 2021). As shown in Fig 1C, CC-Fu$^{EE}$ but not CC-Fu$^{KR}$ phosphorylated the GST-CiC fusion protein containing the wild-type (WT) sequence. Substitution of S1382 (S1382A) or both S1382 and S1385 to Ala (2SA) completely abolished the phosphorylation of corresponding GST-CiC fusion proteins (Fig 1C), suggesting that S1382 is a direct Fu phosphorylation site.

Fu phosphorylates Ci on S218 (N-terminal site) and S1230 (middle site), which primed Ci phosphorylation by CK1 on adjacent sites

(Han et al, 2019). We found that S1382 is followed by T1384 and S1385, both of which fall into the CK1 consensus site: $D/E/(p)S/T[X_{1-3}]$**S/T**, where **S/T** is the phospho-acceptor site and two amino acids (X = 2) provide an optimal spacing between the phospho-acceptor site and $D/E/(p)S/T$ (Knippschild et al, 2005). To determine whether phosphorylation of S1382 by Fu can prime CK1 phosphorylation on adjacent sites such as S1385, we carried out sequential in vitro kinase assay in which GST-CiC$^{WT}$ or GST-CiC$^{S1385A}$ was first incubated with CC-Fu$^{EE}$, followed by treatment with a recombinant CK1. A phospho-Ser specific antibody, pSer, was used to detect phosphorylation on S1382/S1385. As shown in Fig S1, pSer detected equal signals from GST-CiC$^{WT}$ and GST-CiC$^{S1385A}$ when these substrates were incubated with CC-Fu$^{EE}$ alone; however, the pSer signal of GST-CiC$^{WT}$ but not GST-CiC$^{S1385A}$ was further increased after CK1 incubation (Fig S1A). As expected, mutating both S1382 and S1385 to Ala (GST-CiC$^{2SA}$) abolished CiC phosphorylation detected by the pSer antibody in the in vitro kinase assay (Fig S1B). These results suggest that CK1 can phosphorylate S1385 after primed phosphorylation of S1382 by Fu.

To further investigate S1382 phosphorylation, we developed a phospho-specific antibody that recognized phosphorylation on S1382 (pS1382). The in vitro kinase assay described above showed that pS1382 specially recognized GST-CiC$^{WT}$ but not GST-CiC$^{S1382A}$ or GST-CiC$^{2SA}$ after incubation with purified CC-Fu$^{EE}$ (Fig 1D). Next, CC-Fu$^{EE}$ or CC-Fu$^{KR}$ and a non-processed form of Ci (Myc-Ci$^{-PKA}$), which has three PKA sites (S838, S856, and S892) mutated to Ala (Wang et al, 1999), were co-transfected into cultured Drosophila S2R$^+$ cells. Cell extracts were immunoprecipitated with a Myc antibody, followed by Western blot analysis with the pS1382 antibody. We found that pS1382 only detected Myc-Ci$^{-PKA}$ phosphorylation in the presence of CC-Fu$^{EE}$ but not in the presence of CC-Fu$^{KR}$ (Fig 1E). Mutating S1382 to Ala in Myc-Ci$^{-PKA}$ completely abrogated the pS1382 signal (Fig 1E), suggesting that activated Fu could phosphorylate full-length Ci on S1382 in intact cells.

Because Fu can also phosphorylate Ci on S218 and S1230 (Han et al, 2019), we asked whether phosphorylation on individual Fu sites depends on the phosphorylation status of other Fu sites. We co-expressed Myc-Ci$^{-PKA-WT}$, Myc-Ci$^{-PKA-S218AS1230A}$, or Myc-Ci$^{-PKA-S1382A}$ with CC-Fu$^{EE}$ in S2R$^+$ cells and detected the phosphorylation of individual sites with the corresponding phospho-specific antibodies. As showed in Fig 1F, mutating S218 and S1230 to Ala did not affect phosphorylation of pS1382 by Fu. Similarly, mutating S1382 to Ala did not affect phosphorylation of S218 and S1230 by Fu (Fig 1F). Our previous study indicated that phosphorylation on S218 and S1230 was independent of each other (Han et al, 2019). Taken together, these results suggest that phosphorylation on the N-terminal, middle, and C-terminal sites by Fu occurred independently of one another.

To determine whether phosphorylation on S1382 is stimulated by Hh, S2R$^+$ cells transfected with Myc-Ci$^{-PKA}$ were treated with Hh-conditioned medium. We found that Myc-Ci$^{-PKA}$ was phosphorylated on S1382 upon Hh treatment (Fig 1G). Moreover, Hh induced phosphorylation of endogenous Ci on S1382 in Cl8 cells (Fig 1H), which was abolished by Fu RNA interference (RNAi; Fig 1I), suggesting that Hh induces endogenous Ci phosphorylation on S1382 through Fu.

Because S1382 is located within the C-terminal Sufu-binding domain (Fig 1A) (Han et al, 2015), we asked whether phosphorylation of S1382 and adjacent CK1 sites affected Sufu binding. We

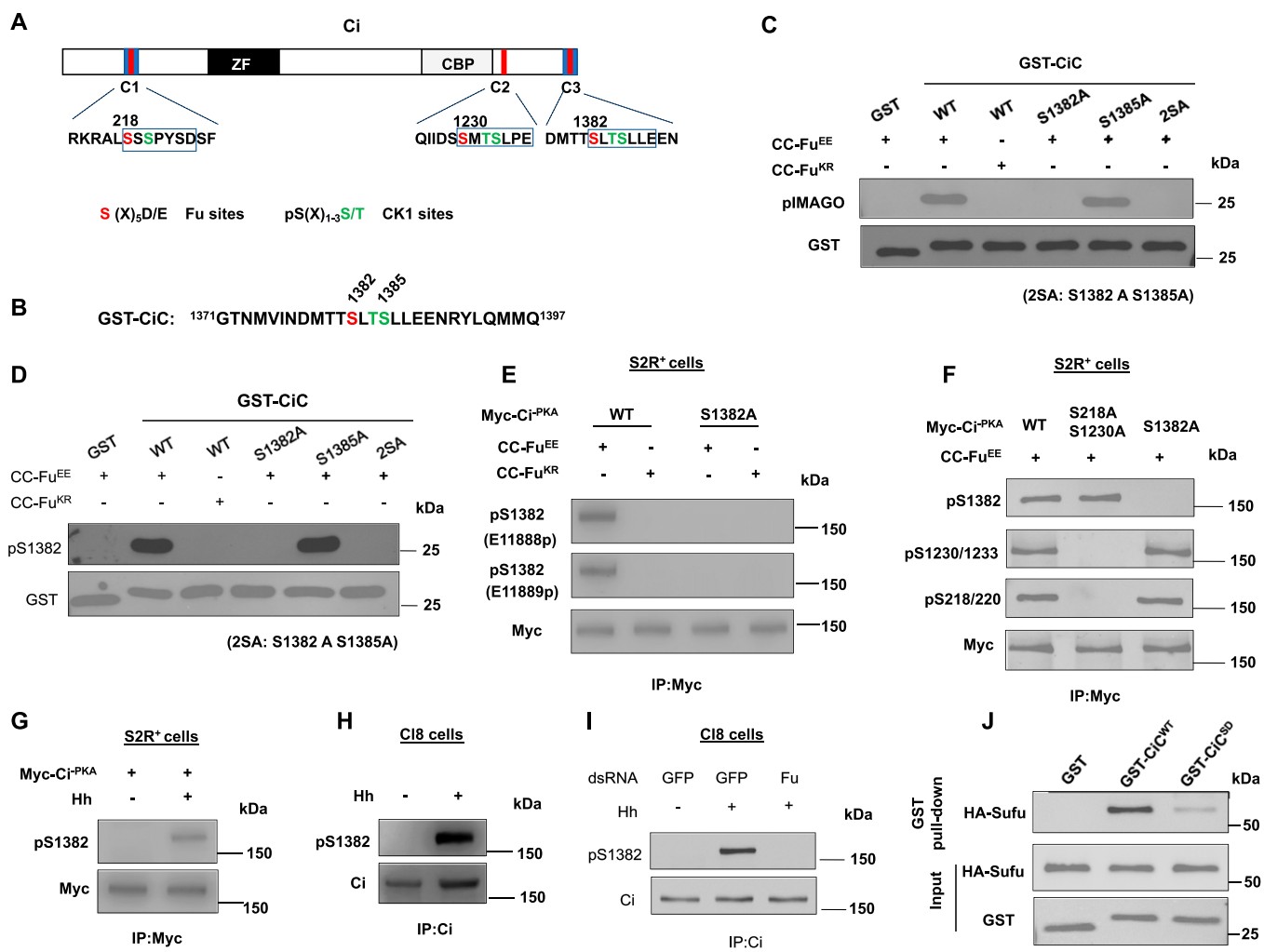

**Figure 1. Fu phosphorylates the C-terminal Sufu-binding domain of Ci.**
**(A)** Schematic diagram of Ci. Blue boxes and red bars indicate the Sufu-binding domains and Fu/CK1 phosphorylation clusters, respectively. ZF, Zinc-Finger DNA-binding domain; CBP, CBP-binding domain. The primary sequences of three phosphorylation clusters are shown with Fu and CK1 sites color-coded in red and green, respectively. **(B)** The primary sequence of the Ci C-terminal region in GST-CiC. **(C, D)** In vitro kinase assay using purified CC-FuEE or CC-FuKR as kinase and the indicated GST-CiC fusion proteins as substrates. Phosphorylation was detected by pIMAGO (C) or pS1382 antibody (D). **(E)** Western blot analysis of Ci phosphorylation in S2R+ cells transfected with the indicated Ci and Fu constructs. Two phospho-specific antibodies (E11888p and E11889p) were used to detect S1382 phosphorylation. **(F)** Western blot analysis of Ci phosphorylation on the indicated sites in S2R+ cells transfected with CC-FuEE and the indicated Myc-Ci-PKA constructs. **(G)** Western blot analysis of Ci phosphorylation on S1382 in S2R+ cells transfected with Myc-Ci-PKA and treated with or without Hh-conditioned medium. **(H)** Western blot analysis of endogenous Ci phosphorylation on S1382 in Cl8 cells treated with or without Hh-conditioned medium. **(I)** Western blot analysis of endogenous Ci phosphorylation on S1382 in Cl8 cells treated with the indicated dsRNA in the absence or presence of Hh. **(J)** Western blot analysis of HA-Sufu pulled down by the indicated GST fusion proteins. 5 µg of GST or GST-CiC fusion proteins were incubated with equal amounts of cell lysates from S2R+ cells transfected with HA-Sufu construct.

generated phospho-mimetic mutations of these sites (S1382D/T1384D/S1385D) in the GST-CiC background (GST-CiCSD). GST pull-down assay showed that GST-CiCSD bound less HA-Sufu derived from S2R+ cell extracts than GST-CiCWT (Fig 1J), suggesting that Fu/CK1-mediated phosphorylation of Ci C-terminal domain attenuates Sufu/Ci interaction.

## Fu-mediated phosphorylation of CiC modulates Ci activity

To determine the relative contribution of individual Fu phosphorylation sites on Ci activation, we mutated the Fu sites individually or in different combinations: SA1 (S218A), SA2 (S1230A), SA3

(S1382A), SA12 (S218A, S1230A), SA13 (S218A, S1382A), SA23 (S1230A, S1382A), and SA123 (S218A, S1230A, S1382A) (Fig 2A). These mutations were generated in the Myc-Ci-PKA background so that the effects of Fu phosphorylation site mutations on Ci activity should reflect changes in the conversion from CiF to CiA (Han et al, 2019). When transfected into S2R+ cells, these Ci variants were expressed at similar levels (Fig S2A). To determine how these mutations affect Fu-mediated Ci activation, we carried out a "de-repression" assay in which coexpression of Sufu with various Ci constructs in S2R+ cells effectively blocked their ability to activate a *ptc-luciferase* (*ptc-luc*) reporter gene, and this repression was alleviated by coexpression of CC-FuEE (Han et al, 2019). As shown in Fig 2B, mutations of

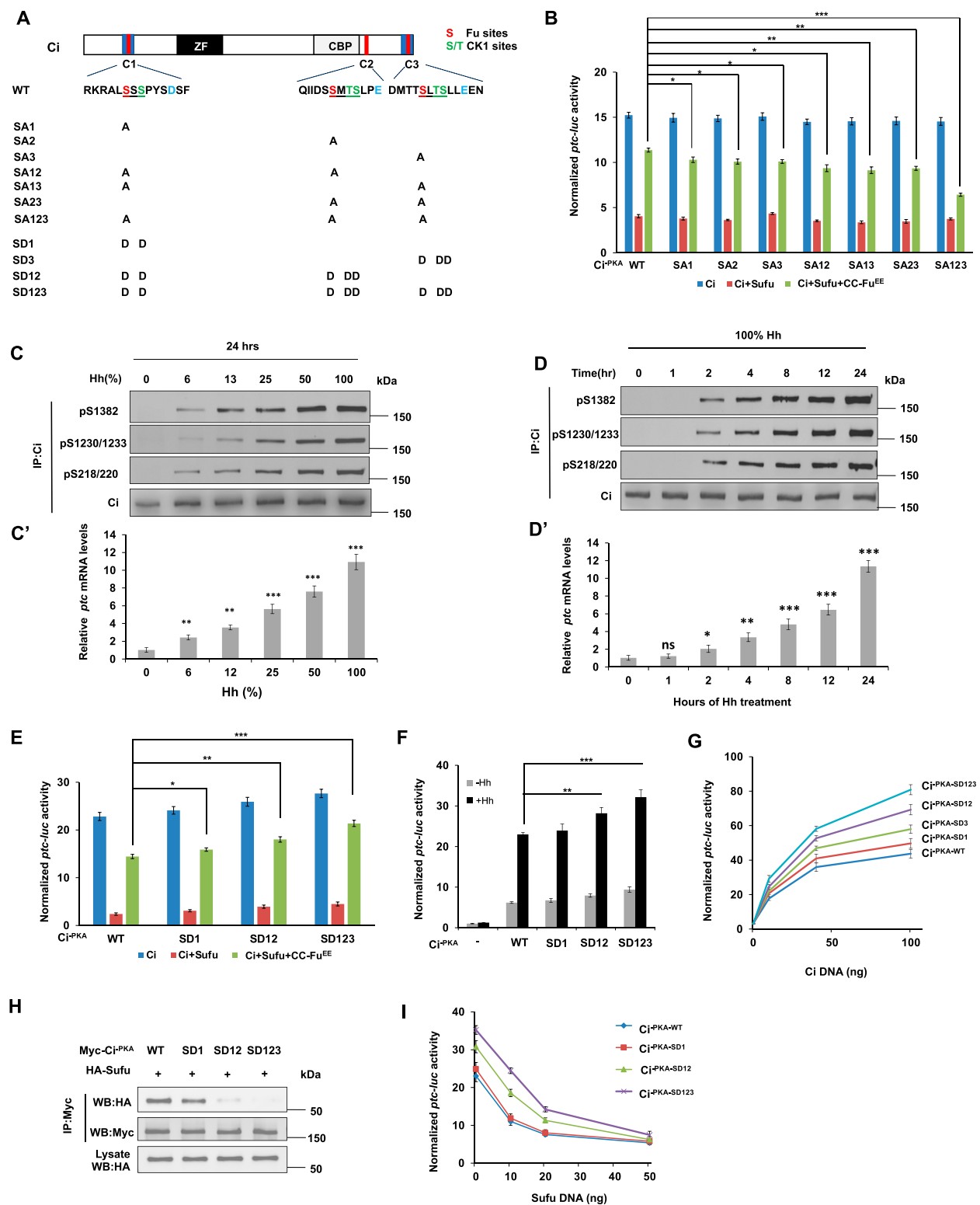

**Figure 2. Dose-dependent phosphorylation and activation of Ci.**
**(A)** Schematic diagram of Ci with primary sequences of the three phosphorylation clusters shown underneath. Fu and CK1 sites are color coded in "red" and "green," respectively. Amino acid substitutions for individual Ci constructs are indicated. **(B)** *ptc-luc* reporter assay in S2R* cells transfected with the indicated Myc-Ci$^{-PKA}$ constructs in the presence or absence of Sufu/CC-Fu$^{EE}$. **(C, C', D, D')** Western blot analyses of endogenous Ci phosphorylation on the indicated sites in Cl8 cells exposed to Hh at increasing concentrations for 24 h (C) or with 100% Hh for increasing amounts of time (D). Hh-conditioned medium was used at a 6:4 dilution ratio with fresh medium (defined as 100%). **(C', D')** Quantification of *ptc* mRNA expression by RT-qPCR in Cl8 cells exposed to Hh at increasing concentrations (C') or duration (D').

individual Fu sites in Ci$^{-PKA}$ only slightly reduced the expression *ptc-luc* compared with wild-type (WT) Ci$^{-PKA}$. Combined mutations of two or more Fu phosphorylation sites resulted in a more dramatic reduction in *ptc-luc* expression with SA123 exhibiting the strongest defect (Fig 2B). Although mutating the C-terminal site alone (SA3) had little effect on Ci activation, this mutation enhanced the defect in Ci activation when combined with other Fu site mutations (Fig 2B). Hence, Fu-mediated phosphorylation of the C-terminal region of Ci modulates Ci activity in conjunction with other phosphorylation events.

## Dose-dependent phosphorylation of Ci in response to the Hh gradient

We next determined how phosphorylation of individual Fu sites are regulated in response to graded Hh signaling. We used the phospho-specific antibodies pS218/220, pS1230/1233, and pS1382 to monitor phosphorylation of endogenous Ci at the N-terminal, middle, and C-terminal Fu sites, respectively, in Cl8 cells treated with conditioned medium containing increasing levels of HhN. We found that phosphorylation detected by pS218/220, pS1230/1233, and pS1382 increased progressively in response to increasing levels of Hh ligand (Fig 2C), which correlated with increasing levels of pathway activity reflected by the levels of *ptc* expression (Fig 2C').

Hh signaling outcomes not only depend on the concentration of Hh ligand but also depend on the duration of Hh ligand exposure (Stamataki et al, 2005). Therefore, we examined how phosphorylation at each Fu site changes over the course of Hh stimulation. Cl8 cells were treated with HhN-conditioned medium for different periods of time and Ci phosphorylation levels at individual sites were monitored by Western blot analyses of immunoprecipitated Ci with the corresponding phospho-specific antibodies. As shown in Fig 2D, phosphorylation at individual Fu phosphorylation sites occurred at about 2 h after Hh exposure, gradually increased over time, and plateaued at 24 h of Hh exposure. The gradual increase in Ci phosphorylation correlated with the progressive increase in *ptc* mRNA expression (Fig 2D'). These results suggest the Ci phosphorylation gradient is determined not only by Hh signal strength but also by signal duration.

## Increasing Fu-mediated phosphorylation progressively increases Ci activity in vitro

To examine whether different levels of Ci phosphorylation determine different levels of Ci activity, we generated a set of phospho-mimetic Ci variants with one, two, or three Fu/CK1 phosphorylation clusters converted to acidic residues (D) to mimic different levels of phosphorylation (Fig 2A): SD1 has cluster 1 mutated to D; SD12 has clusters 1 and 2 mutated to D; and SD123 has all three clusters mutated to D in the Ci$^{-PKA}$ background. When transfected into S2R$^+$

cells, these phospho-mimetic Ci variants were expressed at similar levels (Fig S2B). In the "de-repression" assay, increasing the number of phospho-mimetic mutations progressively increased the *ptc-luc* activity (Fig 2E). When individual Ci variants were expressed at low levels, increasing the number of phospho-mimetic mutations progressively increased both the basal and Hh-induced Ci activity (Fig 2F). In another assay, individual Ci variants were expressed at different levels by transfecting different amounts of DNA constructs and their relative activities were compared at a given expression level. Again, we found that increasing the number of phospho-mimetic mutations resulted in a progressive increase in Ci activity (Fig 2G).

Fu induces the formation of Ci$^A$ by antagonizing Sufu (Ohlmeyer & Kalderon, 1998). In addition, Fu/CK1-mediated phosphorylation of Ci attenuates the binding of Sufu to Ci (Han et al, 2019). Consistent with a gradual change in pathway activity, increasing the number of phospho-mimetic mutations in Ci progressively decreased its binding affinity to Sufu as measured by CoIP experiments (Fig 2H). When fixed amount of WT Myc-Ci$^{-PKA}$ or SD variants were coexpressed with increasing amounts of Sufu, Ci variants with more phospho-mimetic mutations (SD123 and SD12) consistently exhibited higher pathway activity than those with less or no phospho-mimetic mutations (SD1 and WT) before reaching a saturated amount of Sufu which inhibited all the Ci constructs (Fig 2I). These results suggest that increasing levels of Ci phosphorylation by Fu/CK1 render Ci more resistant to Sufu-mediated inhibition.

## Increasing Fu-mediated phosphorylation progressively activates Ci in vivo

To determine how increasing levels of Ci phosphorylation modulate pathway activity in vivo, we generated transgenic flies expressing various Ci constructs under the control of the *UAS* promoter. The *phiC31* integrase system was used to introduce Ci constructs into the same genetic locus to ensure similar expression of different Ci transgenes (Bischof et al, 2007). These Ci transgenes were expressed under the control of a weak Gal4 driver, *C765*, which drives the expression of Ci transgenes at a level close to that of endogenous Ci (Han et al, 2019).

During the development of *Drosophila* wing, cells in the posterior (P) compartment of wing imaginal discs express and secrete Hh, whereas anterior (A) compartment cells express Ci, and consequently, only A-compartment cells near the anterior-posterior (A/P) compartment boundary receive and transduce the Hh signal to activate the Hh pathway target genes such as *ptc* (Fig 3A–A"). When expressed under the control of the *C765* Gal4 driver (*C765>*) that drives the expression of *UAS* transgenes at low levels throughout wing discs (Chen et al, 2010; Ma et al, 2016), both Ci$^{-PKA\_WT}$ and Ci$^{-PKA\_SD1}$ induced ectopic expression of *ptc* only in P-compartment cells but not in A-compartment cells distant from the A/P boundary

---

**(E)** *ptc-luc* reporter assay in S2R$^+$ cells transfected with the indicated Myc-Ci$^{-PKA}$ constructs in the presence or absence of Sufu/CC-Fu$^{EE}$. **(F)** *ptc-luc* reporter assay in S2R$^+$ cells transfected with the indicated Ci constructs at low concentrations (5 ng/well) and treated with (+Hh) or without (−Hh) Hh-conditioned medium. **(G)** *ptc-luc* reporter assay in S2R$^+$ cells transfected with increasing amounts of the indicated Ci constructs. **(H)** Co-immunoprecipitation assay in S2R$^+$ cells transfected with the indicated Myc-Ci$^{-PKA}$ variants and HA-Sufu. **(I)** *ptc-luc* reporter assay in S2R$^+$ cells transfected with a fixed amount of the indicated Ci constructs (50 ng/well) and increasing amounts of Sufu. Data are mean ± SD from three independent experiments. *$P < 0.05$, **$P < 0.01$, and ***$P < 0.001$ (t test).

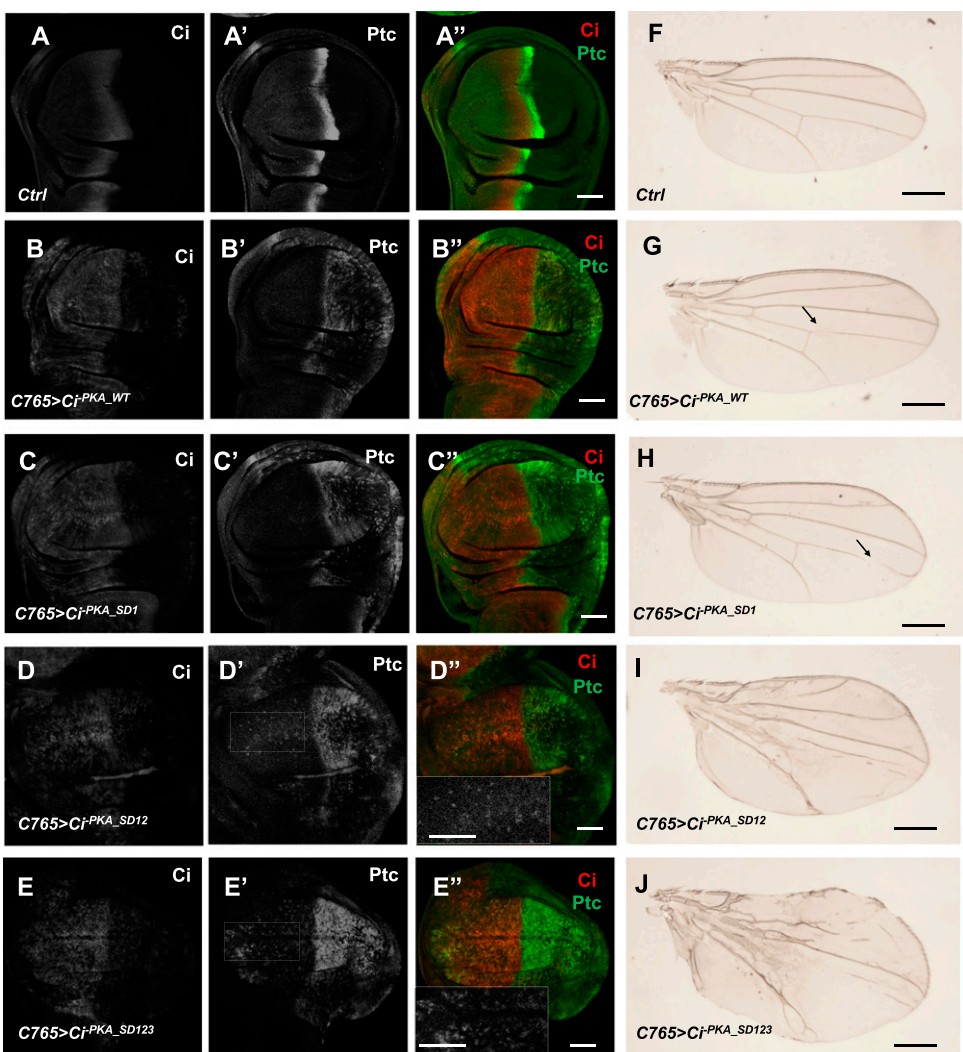

**Figure 3. The effect of increasing phospho-mimetic mutations on Ci activity in vivo.** **(A, B, C, D, E, F, G, H, I, J)** Control (A, A") or wing discs expressing *UAS-Myc-Ci*[-PKA_WT] (B, B"), *UAS-Myc-Ci*[-PKA_SD1] (C, C"), *UAS-Myc-Ci*[-PKA_SD12] (D, D"), or *UAS-Myc-Ci*[-PKA_SD123] (E, E") under the control of *C765* Gal4 driven from larvae grown at 25°C were immunostained to the expression of Ci (white in A, B, C, E; red in A", B", C", D", E") and Ptc (white in A', B', C', D', E'; green in A", B", C", D", E"). Insets in D" and E" are large magnification views of the regions outlined in D' and E', respectively, which demarcate A-compartment cells distant from the A/P boundary. **(F, G, H, I, J)** Adult wings of the indicated genotypes are shown. Arrows in (G) and (H) indicate the gaps on wing vein 4. Scale bars are 50 μM for wing discs and 500 μM for adult wings.

(Fig 3B'–C"), suggesting that neither Ci[-PKA_WT] nor Ci[-PKA_SD1] exhibited Hh-independent pathway activity when expressed at levels close to that of endogenous Ci. Under the same condition, Ci[-PKA_SD12] induced weak ectopic expression of *ptc*, whereas Ci[-PKA_SD123] induced relatively strong ectopic *ptc* expression in A-compartment cells distant from the A/P boundary (Fig 3D'–E"), suggesting that Ci[-PKA_SD123] exhibited higher Hh-independent activity than Ci[-PKA_SD12]. Of note, Ci[-PKA_SD123] activated *ptc* expression more robustly in P-compartment cells than in A-compartment cells distant from the A/P boundary, suggesting that the activity of Ci[-PKA_SD123] is still up-regulated by Hh.

Compared with control wings (Fig 3F), wings expressing Ci[-PKA_WT] or Ci[-PKA_SD1] appear normal in overall morphology with only minor defects such as gaps in wing vein 4 (arrows; Fig 3G and H). By contrast, wings expressing Ci[-PKA_SD12] showed enlarged size and abnormal wing vein patterns (Fig 3I). The overgrowth phenotype and vein patterning defect were exacerbated in wings expressing Ci[-PKA_SD123] (Fig 3J), consistent with Ci[-PKA_SD123] being more potent than Ci[-PKA_SD12] in the transduction of Hh signaling. Hence, the severity of wing patterning defects caused by various Ci mutants

correlated with their relative pathway activities, which were revealed by their abilities to induce ectopic *ptc* expression.

Gal4 activity is temperature-sensitive and drives *UAS* transgene expression at higher levels at 30°C than at room temperature (Brand et al, 1994). To differentiate the activity between Ci[-PKA_WT] and Ci[-PKA_SD] in vivo, larvae expressing *C765>Ci*[-PKA_WT] or *C765>Ci*[-PKA_SD1] were grown at 30°C 3 d before dissection and immunostaining. Under this condition, both Ci[-PKA_WT] and Ci[-PKA_SD1] induced ectopic expression of *ptc* in A-compartment cells distant from the A/P boundary and wing overgrowth, with Ci[-PKA_SD1] causing more severe phenotypes (Fig S3), suggesting that Ci[-PKA_SD1] is more active than Ci[-PKA_WT]. Taken together, these observations suggest that increasing Fu/CK1-mediated phosphorylation progressively activates Ci in vivo.

### C-terminal phosphorylation modulates Gli2 activity

Both Gli2 and Gli3 contain a conserved Sufu-binding domain in their C-terminal region (Han et al, 2015). Sequence alignment between Ci and mouse Gli2/3 identified a similar phosphorylation cluster in the

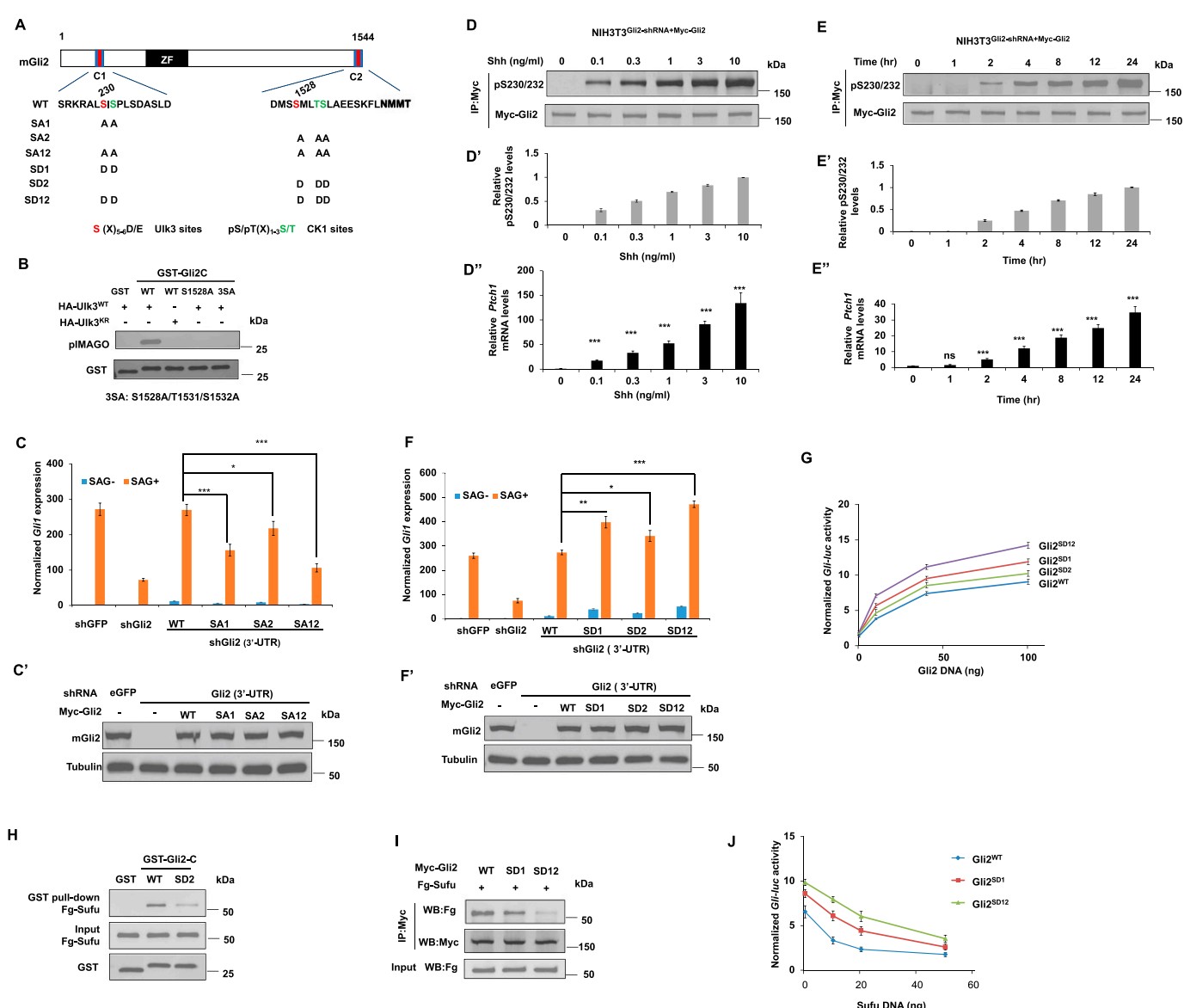

**Figure 4. Multisite phosphorylation of Gli2 promotes its activation.**
**(A)** Schematic diagram of mouse Gli2 with primary sequences of the three phosphorylation clusters shown underneath. Ulk3 and CK1 sites are color-coded in "red" and "green," respectively. Amino acid substitutions for individual Gli2 constructs are indicated. **(B)** In vitro kinase assay using immunopurified HA-Ulk3$^{WT}$ or HA-Ulk3$^{KR}$ as kinase and the indicated GST-fusion proteins as substrates. Phosphorylation was detected by pIMAGO. **(C, C', F, F')** Relative *Gli1* mRNA expression (C, F) or Gli2 protein levels (C' and F') in NIH3T3 cells expressing the indicated shRNA and Myc-Gli2$^{WT}$ or its variants with or without SAG. **(D, D', D", E, E', E")** Western blot analysis of Myc-Gli2 phosphorylation on S230/232 in NIH3T3 cells infected with lentivirus expressing shRNA targeting the 3' UTR of Gli2 and lentivirus expressing Myc-Gli2$^{WT}$ (NIH3T3$^{Gli2-shRNA/Myc-Gli2}$) in the presence of increasing amounts of Shh-N for 24 h (D) or exposed to a fixed amount of Shh-N (1 ng/ml) for increasing amounts of time (E). Quantification of pS230/232 signals (D', E') or *Ptch1* mRNA (D", E") under these conditions. **(G, J)** *Gli-luc* reporter activity in NIH3T3 cells transfected with increasing amounts of the indicated Gli2 constructs (G) or fixed amount of the indicated Gli2 constructs and increasing amounts of Sufu constructs (J). **(H)** Western blot analysis of Fg-Sufu pulled down by the indicated GST fusion proteins from HEK293T cells expressing a Fg-Sufu construct. **(I)** Western blot analysis of Flag-Sufu coimmunoprecipitated with Myc-Gli2$^{WT}$, Myc-Gli2$^{SD1}$, and Myc-Gli2$^{SD12}$. Data are mean ± SD from three independent experiments. *$P < 0.05$, **$P < 0.01$, and ***$P < 0.001$ ($t$ test).

Gli2/3 C-terminal region (Figs 4A and S4). S1528 is a putative Fu/Ulk3 site because there is an acidic residue (E) at +7 position although the previously identified Fu/Ulk3 sites all have an acidic residue at +6 position (Han et al, 2019). We speculated that the spacing between the phospho-acceptor site and the acidic residue could be flexible. Indeed, in vitro kinase assay showed that immunopurified wild-type Ulk3 (HA-Ulk3$^{WT}$) but not a kinase dead form (HA-Ulk3$^{KR}$) phosphorylated a GST fusion protein carrying the Gli2 C-terminal

region (GST-Gli2C; Fig 4B). Substitution of S1528 to Ala or combined mutation of S1528 and two downstream CK1 consensus sites (3SA: S1528A/S1532A/T1531A) abolished the phosphorylation (Fig 4B), suggesting that Ulk3 can directly phosphorylate Gli2 on S1528.

Gli proteins contain a conserved Ulk3/CK1 phosphorylation cluster near their N-terminal Sufu-binding domain and mutating this cluster to Ala in Gli2 (SA1:S230A/S232A) attenuated its activation by Shh in NIH3T3 cells (Han et al, 2019). To determine the

contribution of the C-terminal phosphorylation sites to Gli2 activation, we mutated them to Ala either alone (SA2: S1528A/S1532A/T1531A) or in combination with the N-terminal sites (SA12: SA230A/S232A/S1528A/S1532A/T1531A) (Fig 4A). Lentiviral infection was used to stably express wild type Gli2 (WT) and Gli2 variants (SA1, SA2, and SA12) in NIH3T3 cells with endogenous Gli2 knocked down via an shRNA targeting the 3′ UTR of Gli2 (Han et al, 2019). Western blot analysis confirmed that the exogenously expressed WT Gli2 and SA variants were expressed at levels close to the endogenous Gli2 level in these cell lines (Fig 4C'). As shown in Fig 4C, the C-terminal phosphorylation cluster mutation (SA2) reduced Gli2 activation mediated by the Smo agonist SAG, albeit less dramatically compared with the N-terminal phosphorylation cluster mutation (SA1). Combined mutation of the two clusters (SA12) further reduced SAG-induced Gli2 activation compared with the single-cluster mutations (Fig 4C), suggesting that phosphorylation of Gli2 at both its N- and C-terminal sites contributes to its activation.

Using the phospho-specific antibody that recognizes the N-terminal sites (pS230/S232) (Han et al, 2019), we found that Gli2 phosphorylation at these sites in NIH3T3 cells increased progressively in response to increasing levels of Shh or increasing duration of Shh exposure, which correlated with a progressive increase in Shh pathway activity determined by *Ptch1* mRNA expression (Fig 4D–E"), suggesting that both Shh signal strength and duration determine the levels of Gli2 phosphorylation by Ulk3/CK1.

To determine whether increasing levels of Gli2 phosphorylation by Ulk3/CK1 could progressively increase Gli2 activity, we generated phospho-mimetic mutations in N- (SD1) or C-terminal sites (SD2) either alone or in combination (SD12) (Fig 4A). These Gli2 variants were introduced via lentivirus into NIH3T3 cells with endogenous Gli2 depleted by shRNA targeting the Gli2 3′ UTR. Stable Cell lines expressing Gli2 variants at levels close to that of the endogenous Gli2 were selected for further analysis (Fig 4F'). In agreement with our previous finding (Han et al, 2019), phospho-mimetic mutation of the N-terminal sites (SD1) enhanced SAG-induced Gli2 activation (Fig 4F). We found that phospho-mimetic mutation of the C-terminal sites (SD2) also increased SAG-induced Gli2 activity albeit as lower magnitude compared with SD1 and that combined mutation (SD12) further increased Gli2 activity compared with single-site mutation (Fig 4F). In another assay, Gli2$^{WT}$, Gli2$^{SD1}$, Gli2$^{SD2}$, and Gli2$^{SD12}$ were expressed at increasing levels by transfecting increasing amounts of DNA constructs and their relative activities (measured by *Gli-luc* activity) were compared at a given expression level. As shown in Fig 4G, increasing the number of phospho-mimetic mutations resulted in a progressive increase in Gli2 activity.

By GST pull-down assay, we found that SD2 mutation reduced the binding of Sufu to the C-terminal region of Gli2 (Fig 4H), suggesting that C-terminal phosphorylation of Gli2 interfered with its binding to Sufu. Co-IP experiments showed that increasing the number of phospho-mimetic mutations in Gli2 gradually diminished Sufu binding to full-length Gli2 (Fig 4I), suggesting that increasing levels of Ulk3/CK1–mediated phosphorylation activate Gli2 by gradually alleviating Sufu-mediated repression. In support of this notion, we found that Gli2$^{SD1}$ and Gli2$^{SD12}$ were more resistant to Sufu-mediated inhibition than Gli2$^{WT}$ (Fig 4J).

To determine the effect of Gli2 phosphorylation on its activation in a more physiological context, we turned to an in vitro differentiation assay where mouse embryonic stem cells (mESCs) were induced to differentiate into spinal cord neural progenitor cells that can respond to different levels of Shh and express different Shh target genes that specify progenitor subtypes (Fig 5A and B; Gouti et al, 2014; Pusapati et al, 2018). We depleted endogenous Gli2 from mESCs by shRNA targeting the 3′ UTR of Gli2 and introduced Myc-tagged Gli2$^{WT}$, Gli2$^{SD1}$ and Gli2$^{SD12}$ via lentiviral infection (Fig 5C). mESCs that expressed Gli2$^{WT}$, Gli2$^{SD1}$, or Gli2$^{SD12}$ at similar levels were cultured in N2B27 medium supplemented with bFGF for 2 d and then treated with retinoic acid (RA) alone or RA plus ShhN for 24 h, followed by RT-qPCR to determine the expression of Shh responsive genes (Fig 5A and B). *Foxa2* and *Nkx6.*1 are two Hh target genes that respond to high levels of Shh. In response to Shh stimulation, Gli2$^{SD1}$ variants induced higher expression of *Nkx6.1* and *Foxa2* than Gli2$^{WT}$, and Gli2$^{SD12}$ exhibited the highest activity toward activating these high threshold Shh target genes (Fig 5D and E). On the other hand, Gli2$^{WT}$, Gli2$^{SD1}$, and Gli2$^{SD12}$ activated the expression of a low threshold Shh target gene (*Olig2*) equally well (Fig 5F), consistent with Gli2 being required for high but not low threshold response to Shh (Ding et al, 1998; Matise et al, 1998). These results suggest that the Ulk3/CK1–mediated phosphorylation modulates Gli2 activation during spinal cord neural progenitor cell differentiation.

## Discussion

The Hh morphogen gradient governs cell growth and patterning by generating opposing gradients of Ci$^R$/Gli$^R$ and Ci$^A$/Gli$^A$. It has been well-established that PKA/GSK3/CK1–mediated multisite phosphorylation regulates the formation of Ci$^R$/Gli$^R$. Here, we provided evidence that Fu/Ulk3/CK1–mediated multisite phosphorylation plays a conserved role in the regulation of the Ci$^A$/Gli$^A$ activity gradient. We showed that Hh induces Ci/Gli phosphorylation at conserved Fu/Ulk3/CK1 sites in a dose- and time-dependent manner. Furthermore, we found that increasing the number of phospho-mimetic mutations of the physiologically relevant Fu/Ulk3/CK1 sites progressively increased Ci$^A$/Gli$^A$ activity by gradually alleviating Sufu-mediated inhibition (Fig 6). We propose that the Hh morphogen gradient is translated into a Ci/Gli phosphorylation gradient that contributes to the formation of a Ci$^A$/Gli$^A$ activity gradient.

Previous studies showed that graded Hh signaling in *Drosophila* is translated into Smo phosphorylation and activity gradients, whereby increasing Smo phosphorylation levels progressively increase Smo cell surface accumulation and switch Smo C-terminal intracellular tail (C-tail) from a closed and inactive formation to an open and active formation, leading to dimerization/oligomerization of Smo C-tails (Jia et al, 2004; Zhao et al, 2007; Chen et al, 2010; Li et al, 2016). Smo transduces the Hh signal by directly interacting with Cos2/Fu and inducing Fu dimerization, autophosphorylation, and activation (Jia et al, 2003; Lum et al, 2003; Ogden et al, 2003; Ruel et al, 2003; Shi et al, 2011; Zhang et al, 2011; Zhou & Kalderon, 2011). The Smo gradient is thought to be translated into a Fu activity gradient by inducing increasing levels of its kinase activation loop phosphorylation (Shi et al, 2011). In addition, phosphorylation of Fu regulatory domain may also contribute to its activation (Zhou &

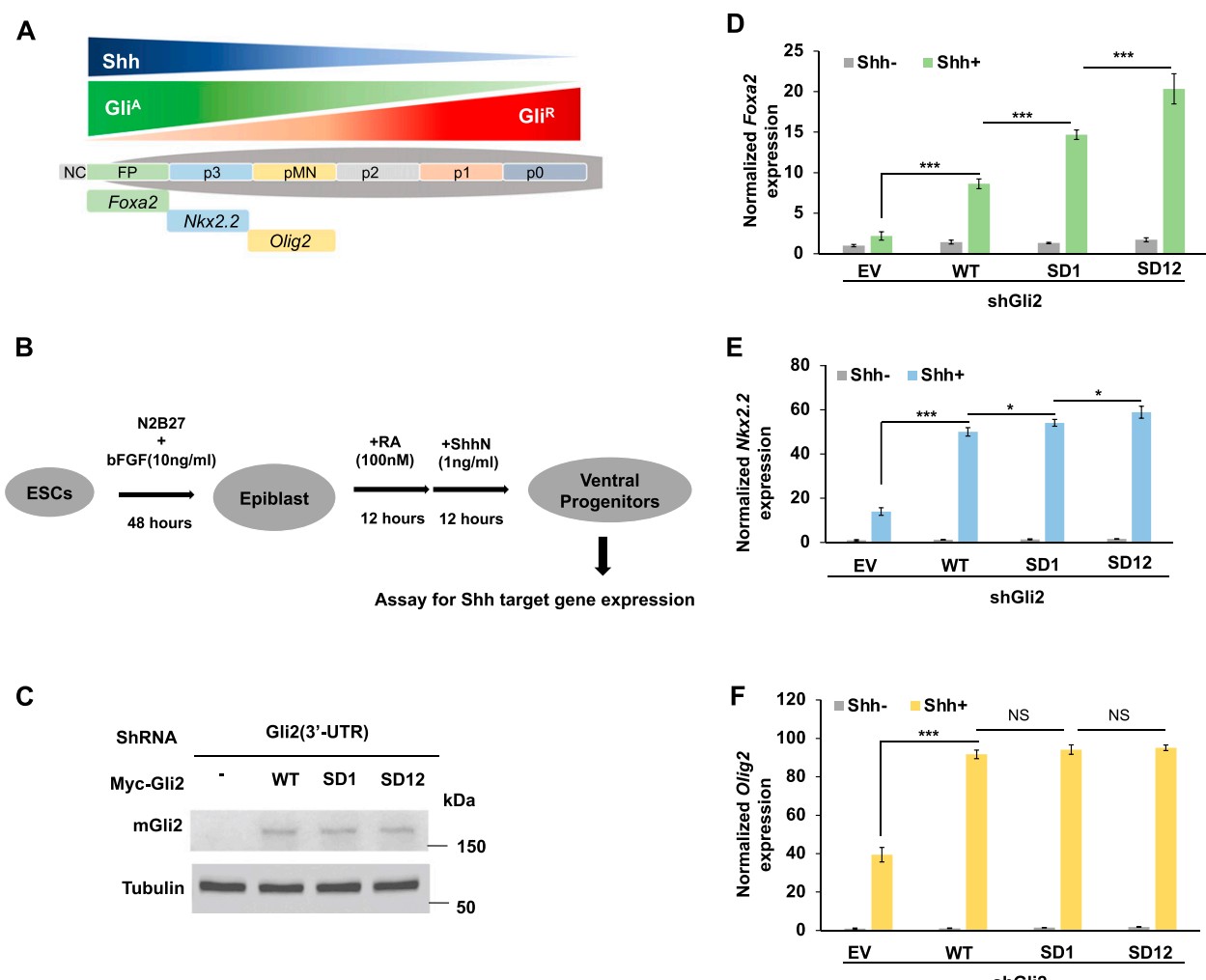

**Figure 5. The effect of increasing phospho-mimetic mutations on Gli2 activity in neural progenitors derived in vitro.**
**(A)** A schematic drawing of ventral patterning of vertebrate neural tube by Shh and Gli[A]/Gli[R] gradients. NC, notochord; FP, floor plate; P3-P0, progenitors for V3-V0 interneurons; pMN, progenitors for motoneurons. **(B)** A flowchart of in vitro differentiation scheme from mouse ES cells to neural progenitor cells. **(C)** Western blot analysis of Gli2 protein from mouse ES cells expressing the indicated shRNA and Myc-Gli2 lentiviral constructs. **(D, E, F)** Relative mRNA levels of *Foxa2* (D), *Nkx2.2* (E), and *Olig2* (F) measured by RT-qPCR in neural progenitor cells derived from mouse ES cells expressing the indicated shRNA and Myc-Gli2 lentiviral constructs with or without Shh-N. Data are mean ± SD from three independent experiments. *$P < 0.05$ and ***$P < 0.001$ ($t$ test). NS, not significant.

Kalderon, 2011). Here, we provided evidence that the Fu activity gradient is further translated into Ci phosphorylation and activity gradients. We found that Ci phosphorylation at multiple Fu sites increased progressively overtime in response to increasing concentrations of Hh, which may reflect the progressively increased Smo and Fu activities (Lum et al, 2003; Fan et al, 2012; Ranieri et al, 2012). Phosphorylation at individual Fu sites on Ci exhibited similar dynamics (Fig 2D), suggesting that phosphorylation of Ci in different regions did not occur in a sequential fashion. Consistent with this notion, mutating one phosphorylation cluster does not appear to affect the phosphorylation on other clusters (Fig 1F) (Han et al, 2019). It is possible that low Hh or short exposure time only resulted in hypophosphorylation of Ci with only one or two Fu/CK1 clusters phosphorylated whereas high Hh or long exposure time resulted in hyperphosphorylation of Ci with all three Fu/CK1 clusters phosphorylated (Fig 6). Alternatively, phosphorylation of the three Fu/

CK1 clusters on individual Ci protein could occur simultaneously and that the number of fully phosphorylated Ci proteins might increase overtime when cells were exposed with a fixed concentration of Hh or in cells treated with increasing concentrations of Hh. At the cellular level, the observed increase in Ci phosphorylation in response to increasing Hh concentration and exposure time could be because of increased responsiveness of individual cells or/and increased number of cells responding to Hh in the population. To address these questions, tools for quantifying Ci phosphorylation at single molecule or cell level need to be developed.

Our previous study identified a phosphorylation consensus sequence for the Fu family of kinases: $S/TX_5D/E$ (Han et al, 2019). However, we speculate that the spacing between the phosphorylation acceptor site and the downstream acidic residual is likely to be flexible. Indeed, the C-terminal site in Gli2 (S1528) has a spacing of six amino acids instead of five (Fig S4). This flexibility may

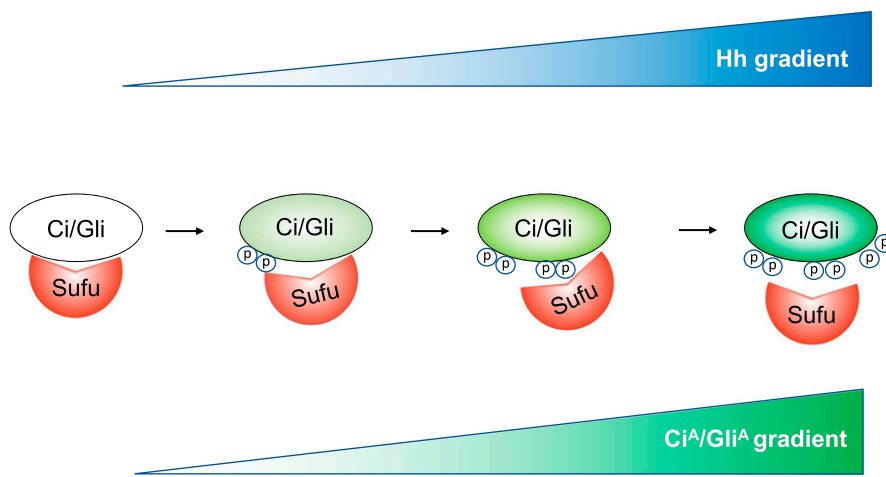

**Figure 6. Hh activates Ci/Gli through differential phosphorylation.**
Graded Hh signals gradually increase the overall levels of Ci/Gli phosphorylation on multiple Fu/Ulk3/CK1 sites, leading to gradual release of Sufu repression and progressive increase in Ci$^A$/Gli$^A$ activity. See text for details.

increase the number of target sites for this family of kinases and can serve as guidance for identifying additional phosphorylation events and physiological relevant substrates regulated by the Fu family of kinases. Of note, the sequence surrounding Gli2 S1528 confers to the CK1 consensus site (Fig S4), raising the possibility that S1528 phosphorylation could also be catalyzed by CK1. Consistent with this, CK1 has been implicated as a positive regulator of both Smo and Ci/Gli2 (Chen et al, 2011; Shi et al, 2014b). We found that phosphorylation at the N-terminal Fu/Ulk3/CK1 cluster in Gli2 increased progressively in response to increasing amounts of Shh or ligand exposure time, but whether phosphorylation at the C-terminal cluster follows the same dynamics remains to be determined.

How does Fu/Ulk3/CK1–mediated phosphorylation of Ci/Gli contribute to Ci/Gli activation? Our previous studies identified multiple Sufu-binding sites in Ci and Gli proteins including Gli2 and Gli3 (Han et al, 2015). Binding of Sufu to Ci/Gli inhibits Ci/Gli by masking its nuclear localization signal and coactivator-binding domain (Shi et al, 2014a; Han et al, 2015). The N- and C-terminal Fu/Ulk3/CK1 phosphorylation clusters are located near the N- and C-terminal Sufu-binding sites, respectively (Figs 1A and 4A). Phospho-mimetic mutations at Fu/Ulk3/CK1 phosphorylation clusters decreased Sufu binding to Ci/Gli and increased the binding of Transportin (Trn) and *Drosophila* CBP (dCBP) to Ci (Figs 1J, 2H, and 4H and I) (Han et al, 2019), suggesting that Fu/Ulk3/CK1–mediated phosphorylation activates Ci/Gli by altering the binding of Sufu and thereby increasing Ci/Gli nuclear translocation and coactivator recruitment. However, it remains possible that Sufu may not completely dissociate from Ci/Gli, and the decreased binding affinity between Sufu and Ci/Gli may reflect a change in the conformation of Ci/Gli-Sufu complex (Zhang et al, 2013). Indeed, Sufu has been shown to accompany activated Ci/Gli to enter the nucleus in response to Hh (Zhang et al, 2017; Roberto et al, 2022). Further study is needed to determine the precise mechanism by which Fu/Ulk3/Stk36 releases Sufu-mediated inhibition of Ci/Gli.

We noticed that Ci$^{-PKA\_SD123}$ was still activated by CC-Fu$^{EE}$ and Hh (Figs 2E and F and 3E′), suggesting that additional mechanisms regulated by Hh/Fu may contribute to Ci/Gli activation. For example, Hh signaling could activate Ci by releasing the inhibition of

full-length Ci by PKA and Cos2 independent of Ci processing (Price & Kalderon, 1999; Wang et al, 1999; Little et al, 2020). Hh signaling could induce Ci phosphorylation by CK1 on multiple HIB/SPOP degrons to protect Ci$^A$ from premature degradation by Cul3$^{HIB/SPOP}$ (Zhang et al, 2006; Shi et al, 2014b). Fu could activate Ci by phosphorylating additional sites on Ci or other substrates. Finally, Fu could allosterically regulate Ci/Sufu interaction independent of its kinase activity. Future work is needed to explore these possibilities.

## Materials and Methods

### DNA constructs

Myc-Ci$^{-PKA}$ was described previously (Han et al, 2019). Myc-Ci$^{-PKA}$ variants with point mutations in different phosphorylation sites shown in Fig 2A were generated using PCR-based site directed mutagenesis. Myc-Ci$^{-PKA}$ and its variants were subclones into the *pUAST* vector (Han et al, 2019). All GST-fusion proteins were subcloned into the *pGEX 4T-1* vector using EcoRI and XhoI sites. N-terminally 6XMyc-tagged mouse Gli2$^{WT}$, Gli2$^{SA1}$, Gli2$^{SA2}$, Gli2$^{SA12}$, Gli2$^{SD1}$, Gli2$^{SD2}$, and Gli2$^{SD12}$ and N-terminally 3XHA-tagged mouse Ulk3$^{WT}$ (HA-Ulk3$^{WT}$) and Ulk3$^{KR}$ (HA-Ulk3$^{KR}$) were subcloned into the *pcDNA3.1*(+) vector using EcoRI and XbaI sites.

### *Drosophila* strains and husbandry

*Drosophila* culture and crosses were carried out according to standard procedure. Transgenic flies expressing Ci constructs (*UAS-Myc-Ci$^{-PKA\_WT}$, UAS-Myc-Ci$^{-PKA\_SD1}$, UAS-Myc-Ci$^{-PKA\_SD12}$*, and *UAS-Myc-Ci$^{-PKA\_SD123}$*) were generated using the *phiC31* integration system (Bischof et al, 2007; Jia et al, 2009). Gal4 driver *C765* (BL#36523) was used to drive the expression of *UAS* transgenes.

### Cell culture, transfection, and lentiviral production

*Drosophila* S2R$^+$ cells were cultured in *Drosophila* serum-free medium (Gibco) containing 10% FBS (Gibco) at 24°C. Clone-8 cells were cultured in Shields and Sang M3 Insect Medium

(Sigma-Aldrich) with 2.5% FBS (Gibco), 2.5% fly extract (1645670, DGRC), insulin (0.125 IU/ml; Sigma-Aldrich) at 24°C. The Calcium Phosphate Transfection Kit (Specialty Media) was used to perform the transfection experiments according to the manufacturer's instruction. The NIH3T3 cells were cultured in DMEM (Sigma-Aldrich) containing 10% bovine calf serum (Gibco) and 1% penicillin/streptomycin (Sigma-Aldrich). The transfection was carried out by using the GenJet Plus in vitro DNA transfection kit (SignaGen). HEK293T cells were cultured in DMEM (Sigma-Aldrich) supplemented with 10% FBS (Gibco) and 1% penicillin/streptomycin (Sigma-Aldrich). The PolyJet in vitro DNA transfection kit (SignaGen) was used to do the transfection according to the manufacturer's instruction. For lentivirus production, the N-terminal 6× Myc-tagged mouse WT and Gli2$^{SD12}$ (Myc-Gli2$^{SD12}$) and Gli2$^{SA12}$ (Myc-Gli2$^{SA12}$) were subcloned into the lentiviral vector FUGW. The expression vectors and package vectors (PSPAX2 and PMDG2) were co-transfected into HEK293T cells using PolyJet (SignaGen). After 48 h, the recombinant viruses were harvested for cell line infection. The recombinant viruses were infected into the NIH3T3 cells using the standard method.

HhN-conditioned medium treatment was performed as previously described (Han et al, 2019). Briefly, the hygromycin (200 µg/ml) was used to select the S2R$^+$ cells stably expressing the HhN (the N-terminal fragment of Hh) which is responsible for the hedgehog signaling. The cells were cultured in the medium without hygromycin but with 0.7 mM CuSO$_4$ for 1 d. Then the medium was harvested and sterilized by filtration. Hh-conditioned medium was used at a 6:4 dilution ratio with fresh medium (defined as 100% in Fig 2C and D). For Shh or SAG treatment, cells were starved for 12 h (0.5% BCS), recombinant human Shh N-terminal fragment (1 ng/ml, #8908-SH-005; R&D Systems) or SAG (200 nm, Cat. no. 566660; Sigma-Aldrich) was added to the medium for 12 h or overnight.

## Immunostaining, immunoprecipitation, and Western blotting analysis

Immunostaining of imaginal discs was carried out as previously described (Jiang & Struhl, 1995). Immunofluorescence imaging was performed using a Zeiss 710 laser scanning microscope. For immunoprecipitation assay, after transfection for 48 h, cells were washed twice with PBS and then lysed on ice for 10 min with lysis buffer containing 1M Tris, pH 8.0, 5M NaCl, 1MNaF, 0.1M Na3VO4, 1% CA630, 10% glycerol, and 0.5M EDTA (pH 8.0). Cell lysates were incubated with protein A-Sepharose beads (Thermo Fisher Scientific) for 1 h at 4°C to eliminate nonspecific binding proteins. After removal of the protein-A beads by centrifugation, the cleared lysates were incubated with Myc (HA or Flag) antibody for 2 h or overnight. Protein complexes were collected by incubation with protein A-Sepharose beads for 1 h at 4°C, followed by centrifugation. Immunoprecipitates were washed three times for 5 min each with lysis buffer and were separated on SDS–PAGE. Western blot was carried out using standard protocol. Antibodies used in this study were as follows: mouse anti-Myc (Santa Cruz Biotechnology), mouse anti-Flag (Sigma-Aldrich), mouse anti-HA (Santa Cruz Biotechnology), rat anti-Ci 2A1, and mouse anti-Ptc (Developmental Studies Hybridoma Bank).

## Phospho antibodies generation

pS218/220 and pS1230/1233 phospho-specific antibodies were previously described (Han et al, 2019). The pS1382 antibody was made by Abmart using the following phospho-peptides as antigens: TTS(p)LTSLLEE. The phospho-specific antibodies were purified by positive and negative selection using the affinity column conjugated with the corresponding phosphorylated and non-phosphorylated peptides sequentially.

## In vitro kinase assay

In vitro kinase assay was performed by incubating 25 µl of reaction mixtures containing 150 mM Tris–HCl (pH 7.5), 0.2 mM Mg2+/ATP, 1 µg of purified GST-Ci/Gli2 fusion proteins, together with appropriate amount of kinase for 30 min at 30°C. The reaction was terminated by adding 2× SDS loading buffer. The resultant samples were load on SDS–PAGE and subjected to the pIMAGO phospho-protein detection kit with the fluor-680 (Sigma-Aldrich) detection kit and LI-COR Odyssey platform for Western blot analyses of phosphorylated proteins (Millipore Sigma). Flag-CC-Fu$^{EE}$ and Flag-CC-Fu$^{KR}$ were purified from insect cells as described previously (Han & Jiang, 2021). HA-Ulk3$^{WT}$ and HA-Ulk3$^{KR}$ were immunopurified from HEK293 cells transfected with the corresponding constructs. Recombinant CK1 was purchased from Abcam (ab102102).

## Luciferase reporter assay and RNAi

The Dual-Luciferase Reporter Assay System (Promega) was applied to detect the dual–luciferase activity according to the manufacturer's instructions. Briefly, S2R$^+$ cells were cultured in 12-well plates, and on the second day, each well was transfected with 0.5 µg *ptc-luc* reporter construct, 25 ng *RL-PolIII* Renilla construct, and 0.5 µg Ci construct. Samples were collected after 48-h incubation. For Hh treatment, 1 d after the transfection, two-thirds of the medium was replaced with HhN-conditioned medium, and cells were incubated for 24 h before harvest. Each sample was measured in triplicate by using a FLUOstar OPTIMA plate reader (BMG Labtech). The *Gli-luc* reporter assay was carried out by transfecting the NIH3T3 cells in 12-well plate with 0.4 mg *8XGli-Luc* reporter, 0.1 mg *PRL-SV40* Renilla vector, and 0.5 mg Gli2 construct. The luciferase activity of each sample was measured in triplicate by using a FLUOstar OPTIMA plate reader (BMG Labtech). dsRNAs were generated using the MEGAscript High Yield Transcription Kit (Ambion). The dsRNA targeting the coding sequence of GFP was used as a negative control. Fu dsRNA targets the coding sequence between amino acids 324–434. For RNAi experiments, cells were treated with corresponding dsRNA in serum-free culture medium for 8 h at 24°C before switching to regular growth medium. The dsRNA-treated cells were cultured for 2 d before analysis. shRNA plasmids against eGFP (control, SHC005) and mGli2 (TRCN0000219066) were purchased from Sigma-Aldrich.

## Quantitative RT-PCR

Total RNA was extracted from 1 × 10$^6$ cells using the RNeasy Plus Mini Kit (QIAGEN), and cDNA was synthesized with the High-Capacity

cDNA Reverse Transcription Kit (Applied Biosystems), and qPCR was performed using Fast SYBR Green Master Mix (Applied Biosystems) and a Bio-Rad CFX96 real-time PCR system. qRT-PCR was performed in triplicate for each of three independent biological replicates. Quantification of mRNA levels was calculated using the comparative CT method. Primers: Ptc, 5′-ATGGACCGCGACAGCCTCCCA-3′ and 5′-CGACGCAGAAGGTGCTCAGCA-3′; Gli1, 5′-GTGCACGTTTGAAGGCTGTC-3′ and 5′-GAGTGGGTCCGATTCTGGTG-3′; Ptch1, 5′-GAAGCCACA-GAAAACCCTGTC-3′ and 5′-GCCGCAAGCCTTCTCTACG-3′; Foxa2, 5′-GGAGTGTACTCCAGGCCTATTA-3′ and 5′-CTCCACTCAGCCTCTCATTTC-3′; Nkx2.2, 5′-CAGCCTCATCCGTCTCAC-3′ and 5′-TCACCTCCATACCTTTCTCC-3′; GAPDH, 5′-GTGGTGAAGCAGGCATCTGA-3′ and 5′-GCCATG-TAGGCCATGAGGTC-3′.

## Neuron progenitor differentiation

A stable cell line derived from HM1 mESCs harboring 8XGBS-Venus reporter at hypoxanthine-guanine phosphoribosyl transferase (HPRT) locus was first transduced by shRNA lentivirus targeting 3′ UTR of Gli2. The Knock down efficiency was validated by qRT-PCR (>90%). Myc-tagged WT and mutant mGli2 were introduced by lentivirus and further G418 selection. The resultant mESCs were cultured with feeder MEFs onto dishes pre-coated with 0.1% gelatin in medium (DMEM, 15% Optima FBS, 1× MEM nonessential amino acids, 2 mM L-glutamine, 1% Embryo Max nucleosides, 55 $\mu$M 2-mercaptoethanol, and 1,000 U/ml ESGRO LIF). For differentiation, feeder cells were eliminated by trypsinization followed by 2× 10 min incubation for settlement. mESC were then cultured in CellBIND plates at the density of 60,000 cells per six well in N2B27 medium (DMEM F12 and neurobasal medium at 1:1 ratio, N2 and B27 supplement, 2 mM L-glutamine 40 $\mu$g/ml BSA, 55 $\mu$M 2-mercaptoethanol). For the first 2 d (day 0 and 1) after the seeding, cells were cultured in N2B27 medium supplemented with 10 ng/ml bFGF. On day 2, the differentiation was induced by adding RA (100 nM; Sigma-Aldrich) for 12 h. After that, the cells were treated with Shh N (1 ng/ml, R&D System) for another 12 h. Then samples were collected for qRT-PCR.

# Supplementary Information

# Acknowledgements

We thank Dr. Xiaochun Li and Dr. James Briscoe for reagents, the Developmental Studies Hybridoma Bank for antibodies, and the Bloomington Drosophila Stock Center for fly stocks. This work was supported by grants from NIGMS (R35GM118063) and Welch Foundation (I-1603) to J Jiang. J Jiang is a Eugene McDermott Endowed Scholar in Biomedical Science at UTSW.

## Author Contributions

M Zhou: data curation, formal analysis, investigation, and writing—review and editing.

Y Han: conceptualization, data curation, formal analysis, investigation, and methodology.

B Wang: resources, investigation, and project administration.

YS Cho: data curation and formal analysis.

J Jiang: conceptualization, formal analysis, supervision, funding acquisition, project administration, and writing—original draft, review, and editing.

## Conflict of Interest Statement

The authors declare that they have no conflict of interest.

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
