## [Reviewer comments · Life Science Alliance]

Life Science Alliance

Dose-dependent phosphorylation and activation of Hh pathway transcription factors

Jin Jiang, Mengmeng Zhou, Yuhog Han, Bing Wang, and Yong Suk Cho

DOI: <https://doi.org/10.26508/lsa.202201570>

Corresponding author(s): Jin Jiang, UTSW

Review Timeline:

Submission Date:	2022-06-20
Editorial Decision:	2022-07-20
Revision Received:	2022-08-17
Editorial Decision:	2022-08-19
Revision Received:	2022-08-23
Accepted:	2022-08-23

Transaction Report:

July 20, 2022

Re: Life Science Alliance manuscript #LSA-2022-01570-T

Prof. Jin Jiang
UT Southwestern Medical Center
Department of Molecular Biology
Room 5.604B
Dallas, TX 75390

Dear Dr. Jiang,

Thank you for submitting your manuscript entitled "Dose-dependent phosphorylation and activation of Hh pathway transcription factors" to Life Science Alliance. The manuscript was assessed by expert reviewers, whose comments are appended to this letter. We invite you to submit a revised manuscript addressing the Reviewer comments.

Thank you for this interesting contribution to Life Science Alliance. We are looking forward to receiving your revised manuscript.

Sincerely,

B. MANUSCRIPT ORGANIZATION AND FORMATTING:

Reviewer #1 (Comments to the Authors (Required)):

In the submitted work Zhou, Han et al characterize the consequences of Fu phosphorylating the C-terminal Sufu-binding domain of Ci/Gli2. This is done using a combination of biochemical and genetic approaches. The work is very much an extension of Han et al 2019, also from the Jin Jiang lab. The key findings of that paper were: Hh activates Ci by stimulating its phosphorylation through Fu and Fu directly phosphorylates Ci and primes its further phosphorylation by CK1. The data in the submitted manuscript show that the C-terminal motif exhibits the same behavior. Experiments using different combinations of mutants suggest that all the Fu phosphorylation sites need to be taken into account in experiments that want to further elucidate how graded Hh signaling is achieved. Thus, while only a detail, the detail reported in the submitted manuscript maybe key to finishing this puzzle. Neither discussed nor investigated are how the phosphorylation affects Trn and CBP recruitment to Ci (Han et al., 2019), the nuclear (C-term) vs cytoplasmic (N-term) roles of Sufu inhibition (Han et al., 2015,), as well as the potential interaction with Cos2/Fused activation of Ci. Some discussion of these issues would strengthen the manuscript.

While the presented data support the claims that are made some additional experiments and/or discussion would be very beneficial. This criticism is expanded on below in a Figure by Figure manner. Minor comments are also included and indicated.

Figure 1. and EV1 Fu phosphorylates the C-terminal Sufu-binding domain of Ci

Two minor comments.

Figure 1F there is an error in the labelling it reads S281 not S218.

Figure EV1 It would be interesting to see if pSer staining is lost in the 2SA mutant.

Figure 2. Dose-dependent phosphorylation and activation of Ci

Minor comment: Figure 2I not in the Figure Legend

Page 8 Paragraph 1

At the end of the paragraph the claim is made that "Fu-mediated phosphorylation of the C-terminal region of Ci modulates Ci activity." Given the effect is very minor unless combined with other Fu site mutations I would suggest that authors moderate the claim to reflect this. E.g. modulates in concert with the other motifs As an alternative in vivo experiments (as per Fig 3) with a SD3 mutant would buttress the author's claim.

Figure 2B/2E

Fu should not be able to act on the SA123 and SD123 mutants, but there is clearly still an effect. This must be discussed.

Do the authors have western blots to confirm that the different constructs are equally expressed. Given the mild effects that are seen, trivial explanations like differences in expression levels need to be ruled out.

Figure 2C/C'/D/D' To facilitate reproducibility, it would be helpful to indicate in the figure legend at what time point the Hh-titration experiments were terminated and measured. It would also be helpful to know that concentration of Hh was used in the duration experiment. Were different Hh concentrations also tried?

Are there differences in the kinetics/sensitivity to Hh among the different Fu/CK1 clusters that could contribute to how a graded response is readout? Do the authors think all the clusters are equally important?

Figure 3. The effect of phospho-mimetic mutations on Ci activity in vivo

Pages 10 and 11. Figure 3 is incorrectly cited as Figure 2 in many places.

The LSA audience is broad therefore it would aid interpretation if the expression pattern of C765-Gal4 was shown/stated.

In Figure 3E "Ci-PKA_SD123 induced relatively strong ptc expression in A-compartment cells distant from the A/P boundary".

Why only in distant cells and not the entire pouch? C765-Gal4 is expressed at low levels in the entire pouch. Should not the triple-mutations confer some Hh-ligand independence?

It is difficult to visually assess the changes to Ptc expression in the different mutants and it would be clearer if quantifications were included.

Ptc lacZ is taken as a proxy for Hh signaling output. Given the disruption of the Ptc-lacZ pattern, and by extension Hh signaling, it may be useful to explain briefly why wing is pattern in normal.

"Ci transgenes were expressed by a weak Gal4 driver, C765, which drives the expression of Ci transgenes at a level close to that of endogenous Ci"- The Figure 3 panels suggest that Ci is being expressed at normal levels, however it is unclear how this is monitored, (A appears to be 2A1 staining, while B-E are Myc staining). If this is the case, Myc staining can be quite weak so there should be an additional 2A1 staining to confirm similar expression levels. Otherwise this statement is not accurate.

Figure 4. Multisite phosphorylation promotes Gli2 activation

Figure 4D/E: The previously reported Gli2 sites S230/232 are convincingly shown to be phosphorylated by activation of the Hh signaling pathway. However, the main point of this paper is the new C-term site (1528) and this was not tested. If the comparison between Ci/Gli regulation at the C-term is to be supported, then an analogous experiment should be performed.

Figure 4G it would be informative to see the effect of titrating SD2 mutant alone. This would support the claim that this site plays a role in regulating Gli2 activity. By extension the SD3 mutation alone would also be informative to include in Figure 2G.

Figure 4L/M why are only high threshold targets looked at? A central claim is that the discovery helps explain graded signaling activity but readouts of graded signaling (low/med/high threshold targets) are not thoroughly examined.

Discussion

The discussion is rather short and could benefit from some reflection on how the proposed Ci/Gli phosphorylation gradient is combined with other mechanisms for generating a Hh activity gradient, for example Smo phosphorylation. In addition, it would be useful to discuss how the findings complement Han et al., 2015 and Han et al 2019.

Much is made of the fact that phosphorylation increased in response to increasing levels of Hh or increasing amounts of Hh exposure time. It could be helpful to explain why this is unexpected. Naively the observed accumulation (phosphor-Ci, ptc mRNA) is an expected consequence of prolonged Fu activity. How does the used 24 hour time frame correspond to in vivo signaling times and how do the signaling strengths compare.

General comment

There are a number of trivial errors that detract from the quality of the paper (mislabelled figures, typos). For example, the last sentence of the Abstract: "Our study suggests that Hh signaling gradient is translated into a Ci/Gli phosphorylation gradient that activates Ci/Gli by gradually resealing Sufu inhibition."

Reviewer #2 (Comments to the Authors (Required)):

This manuscript describes convincing and well-controlled data showing phosphorylation of Ci and its Gli2,3 homologs that is Hh-dependent and influences both Sufu binding and activation function. These findings are important contributions to understanding how Hh signal transduction is effected, and I strongly recommend its publication. My suggestions are minor:

Abstract: "Our study suggests that Hh signaling gradient is translated into a Ci/Gli phosphorylation gradient that activates Ci/Gli by gradually resealing Sufu inhibition." I don't understand meaning of "resealing". Perhaps "reducing" is intended?

p9 The authors might comment on the possibility that the observed phosphorylation increases with Hh concentration and time might either be a feature of most cells in the population, or might be a consequence of increasing number of cells responding.

p11, 12 check spellings: residual - residue; acid - acidic; phosphorylates - phosphorylate, and more

Dear Dr. Sawey

Enclosed please find our revised manuscript entitled “**Dose-dependent phosphorylation and activation of Hh pathway transcription factors**” for submission to Life Science Alliance. In the revision, we have included additional experiments and discussions suggested by the reviewers (see point-to-point-responses to reviewers' comments below for details). We believe that we have satisfactorily addressed all the reviewers' comments. We thank both reviewers for their constructive comments that help us improve our manuscript. I look forward to your favorable decision.

Sincerely,

Jin Jiang

Reviewer #1 (Comments to the Authors (Required)):

In the submitted work Zhou, Han et al characterize the consequences of Fu phosphorylating the C-terminal Sufu-binding domain of Ci/Gli2. This is done using a combination of biochemical and genetic approaches. The work is very much an extension of Han et al 2019, also from the Jin Jiang lab. The key findings of that paper were: Hh activates Ci by stimulating its phosphorylation through Fu and Fu directly phosphorylates Ci and primes its further phosphorylation by CK1. The data in the submitted manuscript show that the C-terminal motif exhibits the same behavior. Experiments using different combinations of mutants suggest that all the Fu phosphorylation sites need to be taken into account in experiments that want to further elucidate how graded Hh signaling is achieved. Thus, while only a detail, the detail reported in the submitted manuscript maybe key to finishing this puzzle. Neither discussed nor investigated are how the phosphorylation affects Trn and CBP recruitment to Ci (Han et al., 2019), the nuclear (C-term) vs cytoplasmic (N-term) roles of Sufu inhibition (Han et al., 2015,) as well as the potential interaction with Cos2/Fused activation of Ci. Some discussion of these issues would strengthen the manuscript.

While the presented data support the claims that are made, some additional experiments and/or discussion would be very beneficial. This criticism is expanded on below in a Figure by Figure manner. Minor comments are also included and indicated.

We have included additional experiments and discussions suggested by the reviewer.

Figure 1. and EV1 Fu phosphorylates the C-terminal Sufu-binding domain of Ci
Two minor comments.

1. Figure 1F there is an error in the labelling it reads S281 not S218.

We have corrected this mislabelling.

2. Figure EV1 It would be interesting to see if pSer staining is lost in the 2SA mutant.

we carried out *in vitro* kinase assay using GST-CiC^{2SA} as a substrate. As expected, pSer signal was lost with the 2SA mutant (see revised Fig. S1B).

Figure 2. Dose-dependent phosphorylation and activation of Ci

1. Minor comment: Figure 2I not in the Figure Legend

Legends for Fig 2G and 2I were combined in the previous version. We have now separated them in the revision.

2. Page 8 Paragraph 1

At the end of the paragraph the claim is made that "Fu-mediated phosphorylation of the C-terminal region of Ci modulates Ci activity." Given the effect is very minor unless combined with other Fu site mutations I would suggest that authors moderate the claim to reflect this. E.g. modulates in concert with the other motifs As an alternative *in vivo* experiments (as per Fig 3) with a SD3 mutant would buttress the author's claim.

As suggested by the review, we modified this statement as "Fu-mediated phosphorylation of the C-terminal region of Ci modulates Ci activity in conjunction with other phosphorylation events".

3. Figure 2B/2E

Fu should not be able to act on the SA123 and SD123 mutants, but there is clearly still an effect. This must be discussed.

We discussed the possibility that other events beside the three identified clusters on Ci could contribute Hh/Fu-mediated activation of SD123 (the last paragraph of the "Discussion").

Do the authors have western blots to confirm that the different constructs are equally expressed. Given the mild effects that are seen, trivial explanations like differences in expression levels need to be ruled out.

We included western blot analysis of various Ci constructs expressed in S2R⁺ cells in revised Fig. S2 and confirmed that they were expressed at similar levels.

4. Figure 2C/C'/D/D' To facilitate reproducibility, it would be helpful to indicate in the figure legend at what time point the Hh-titration experiments were terminated and measured. It would also be helpful to know that concentration of Hh was used in the duration experiment. Were different Hh concentrations also tried?

Sorry for missing the detailed information. In Fig. 2C/C', we treated cells with different concentration for 24 hours and in figure 2D/D', we treated cells with 100% Hh for different time.

Hh-conditioned medium was used at a 6:4 dilution ratio with fresh medium (defined as 100% in Fig. 2C and D). We have now included this information in the figure and method.

5. Are there differences in the kinetics/sensitivity to Hh among the different Fu/CK1 clusters that could contribute to how a graded response is readout? Do the authors think all the clusters are equally important?

The results shown in Fig. 2C/C'/D/D' did not reveal any obvious differences in the kinetics/sensitivity to Hh among the different Fu/CK1 clusters. The results in Fig. 2 did not reveal striking differences among individual Fu-mediated phosphorylation clusters in their relative contributions to Ci activation. That said, it is still possible that different sites may modulate Ci activity in different ways. e.g., the C-terminal sites could preferentially regulate CBP binding while the N-terminal sites could contribute more to the regulation of Ci nuclear import.

Figure 3. The effect of phospho-mimetic mutations on Ci activity in vivo

1. Pages 10 and 11. Figure 3 is incorrectly cited as Figure 2 in many places.

We have corrected these mistakes. Thanks.

2. The LSA audience is broad therefore it would aid interpretation if the expression pattern of C765-Gal4 was shown/stated.

We stated that *C765-Gal4* drives the expression of *UAS* transgenes at low levels throughout wing discs and provided the references.

3. In Figure 3E "Ci-PKA_SD123 induced relatively strong *ptc* expression in A-compartment cells distant from the A/P boundary". Why only in distant cells and not the entire pouch? C765-Gal4 is expressed at low levels in the entire pouch. Should not the triple-mutations confer some Hh-ligand independence? It is difficult to visually assess the changes to Ptc expression in the different mutants and it would be clearer if quantifications were included.

To improve the visibility of weak staining, we converted the single channel images into black and white images (3A-E for Ci and 3A'-E' for Ptc). We also included insets in 3D" and 3E" to show enlarged views of the outlined regions in 3D' and 3E'. As one can see in the revised Fig. 3E', Ci^{-PKA-SD123} does induce ectopic *ptc* expression in most of the regions in the A compartment, suggesting that SD123 does confer some Hh-independent activity. SD123 drives relatively stronger ectopic *ptc* expression than SD12, which is better shown in the enlarged images.

4. Ptc lacZ is taken as a proxy for Hh signaling output. Given the disruption of the Ptc-lacZ pattern, and by extension Hh signaling, it may be useful to explain briefly why wing is pattern in normal.

We include a statement “the severity of wing patterning defects caused by various Ci mutants correlated with their pathway activities, which were revealed by their abilities to induce ectopic *ptc* expression”.

5. "Ci transgenes were expressed by a weak Gal4 driver, C765, which drives the expression of Ci transgenes at a level close to that of endogenous Ci"- The Figure 3 panels suggest that Ci is being expressed at normal levels, however it is unclear how this is monitored, (A appears to be 2A1 staining, while B-E are Myc staining). If this is the case, Myc staining can be quite weak so there should be an additional 2A1 staining to confirm similar expression levels. Otherwise this statement is not accurate.

All panels (Fig. 3A-E) were stained with the Ci (2A1) antibody. 3B-3E showed superimposed endogenous and exogenous Ci as evidenced by the higher levels of Ci in A compartments (endogenous + exogenous Ci) than in P compartments (exogenous Ci only).

Figure 4. Multisite phosphorylation promotes Gli2 activation

1. Figure 4D/E: The previously reported Gli2 sites S230/232 are convincingly shown to be phosphorylated by activation of the Hh signaling pathway. However, the main point of this paper is the new C-term site (1528) and this was not tested. If the comparison between Ci/Gli regulation at the C-term is to be supported, then an analogous experiment should be performed.

We do not have a phosphor-specific antibody for Gli2 S1528. We tried but failed. Therefore, we can only rely on the *in vitro* kinase assay to show that purified Ulk3^{WT} not Ulk3^{KR} phosphorylated GST-Gli2^{WT} but not GST-Gli2^{S1528A} (Fig. 4B).

2. Figure 4G it would be informative to see the effect of titrating SD2 mutant alone. This would support the claim that this site plays a role in regulating Gli2 activity. By extension the SD3 mutation alone would also be informative to include in Figure 2G.

We included the results on Gli2^{SD2} in the revised Fig. 4G and Ci^{SD3} in the revised Fig. 2G

3. Figure 4L/M why are only high threshold targets looked at? A central claim is that the discovery helps explain graded signaling activity but readouts of graded signaling (low/med/high threshold targets) are not thoroughly examined.

Previous studies showed that Gli2 is only required for the expression of high threshold targets while the low and intermediate threshold targets are mainly regulated by the Gli^R gradient. We now included a low threshold target *Olig2*. As expected, we did not observe significant difference between Gli2SD1 and SD12 in activating *Olig2* (revised fig. 5F).

Discussion

1. The discussion is rather short and could benefit from some reflection on how the proposed Ci/Gli phosphorylation gradient is combined with other mechanisms for generating a Hh activity gradient, for example Smo phosphorylation. In addition, it would be useful to discuss how the findings complement Han et al., 2015 and Han et al 2019.

The original submission was for a short article with "Results" and "Discussion" combined. In the revision, we added a "Discussion" section that includes many topics that the reviewers suggested.

2. Much is made of the fact that phosphorylation increased in response to increasing levels of Hh or increasing amounts of Hh exposure time. It could be helpful to explain why this is unexpected. Naively the observed accumulation (phosphor-Ci, ptc mRNA) is an expected consequence of prolonged Fu activity. How does the used 24 hour time frame correspond to *in vivo* signaling times and how do the signaling strengths compare.

We did not argue that the graded increase in Ci/Gli phosphorylation is unexpected. We agreed with the reviewer that the graded increase in Ci/Gli phosphorylation may reflect a gradual increase in Smo/Fu activity, which has been included in the discussion. The 24 hours' time frame is arbitrarily defined, which does not necessarily reflect the *in vivo* setting. Similarly, it is hard to compare signaling strengths between *in vitro* and *in vivo* settings.

General comment

There are a number of trivial errors that detract from the quality of the paper (mislabeled figures, typos). For example, the last sentence of the Abstract: "Our study suggests that Hh signaling gradient is translated into a Ci/Gli phosphorylation gradient that activates Ci/Gli by gradually resealing Sufu inhibition."

We have corrected these typos.

Reviewer #2 (Comments to the Authors (Required)):

This manuscript describes convincing and well-controlled data showing phosphorylation of Ci and its Gli2,3 homologs that is Hh-dependent and influences both Sufu binding and activation function. These findings are important contributions to understanding how Hh signal transduction is effected, and I strongly recommend its publication. My suggestions are minor:

1. Abstract: "Our study suggests that Hh signaling gradient is translated into a Ci/Gli phosphorylation gradient that activates Ci/Gli by gradually resealing Sufu inhibition." I don't understand meaning of "resealing". Perhaps "reducing" is intended?

We changed the "resealing" to "releasing".

2. p9 The authors might comment on the possibility that the observed phosphorylation increases with Hh concentration and time might either be a feature of most cells in the population, or might be a consequence of increasing number of cells responding.

Thanks for this suggestion. We have included this in the discussion.

3. p11, 12 check spellings: residual - residue; acid - acidic; phosphorylates - phosphorylate, and more

We have corrected these misspellings.

August 19, 2022

RE: Life Science Alliance Manuscript #LSA-2022-01570-TR

Prof. Jin Jiang
UTSW
Department of Molecular Biology
Room 5.604B
Dallas, TX 75390

Dear Dr. Jiang,

Thank you for submitting your revised manuscript entitled "Dose-dependent phosphorylation and activation of Hh pathway transcription factors". We would be happy to publish your paper in Life Science Alliance pending final revisions necessary to meet our formatting guidelines.

- please upload your main and supplementary figures as single files
- please add your supplementary figure legends to the main manuscript text
- please add ORCID ID for corresponding author-you should have received instructions on how to do so
- please add the Twitter handle of your host institute/organization as well as your own or/and one of the authors in our system
- please add a callout for Figure 5C-F to the main manuscript text

Figure Check:

-Figure 3D and E: It is unclear from the figure what the magnification of the insets are. please include a separate scale bar for the insets and label accordingly.

A. FINAL FILES:

B. MANUSCRIPT ORGANIZATION AND FORMATTING:

Sincerely,

August 23, 2022

RE: Life Science Alliance Manuscript #LSA-2022-01570-TRR

Prof. Jin Jiang
UTSW
Department of Molecular Biology
ND5.136AE, 6000 Harry Hines Blvd. Dallas, TX 75390
Dallas, TX 75390

Dear Dr. Jiang,

Thank you for submitting your Research Article entitled "Dose-dependent phosphorylation and activation of Hh pathway transcription factors". It is a pleasure to let you know that your manuscript is now accepted for publication in Life Science Alliance. Congratulations on this interesting work.

DISTRIBUTION OF MATERIALS:

Again, congratulations on a very nice paper. I hope you found the review process to be constructive and are pleased with how the manuscript was handled editorially. We look forward to future exciting submissions from your lab.

Sincerely,
